# Sustainable Development of Underground Coal Resources in Shallow Groundwater Areas for Environment and Socio-Economic Considerations: A Case Study of Zhangji Coal Mine in China

**DOI:** 10.3390/ijerph20065213

**Published:** 2023-03-16

**Authors:** Ruiya Zhang, Yoginder P. Chugh

**Affiliations:** 1Key Laboratory of Roads and Railway Engineering Safety Control, Shijiazhuang Tiedao University, Ministry of Education, Shijiazhuang 050043, China; zhangruiya@stdu.edu.cn; 2Institute of Land Reclamation and Ecological Restoration, China University of Mining and Technology Beijing, Beijing 100083, China; 3College of Engineering, Computing, Technology and Mathematics, Southern Illinois University, Carbondale, IL 62901, USA

**Keywords:** sustainable resource development, coal and mineral resources, shallow groundwater, dynamic subsidence reclamation, super farmland

## Abstract

Coal resources in China are developed in several regions with shallow groundwater, and large mining-related surface subsidence can have negative impacts on agriculture, land and water resources as well as existing and future socio-economic resources. All these are important for sustainable resource development. Dynamic subsidence reclamation (DSR) planning concepts are evaluated here for another case study with analyses over a 11-year period. In DSR topsoil, subsoil, farming, and water resources management are dynamically synergized concurrent with mining ahead of and behind the projected dynamic subsidence trough. The study area involved mining five longwall faces (and post-mining reclamation) to assess if DSR could have improved both the environment and socio-economic conditions for post-mining land use as compared to using traditional reclamation (TR) and TR-modified (TR(MOD)) approaches. The results show that: (1) Upon final reclamation, farmland area and water resources in DSR and TR (MOD) will have increased by 5.6% and 30.2% as compared to TR. Removing soils ahead of mining before they submerge into water is important for farmland reclamation and long-term economic development. (2) Due to topsoil and subsoil separation and storage in the DSR plan, reclaimed farmland productivity should recover quickly and agriculture production should be larger than TR and TR(MOD) plans. (3) For a simplified economic model, the total revenue in the DSR plan should be 2.8 times more than in TR and 1.2 times larger than in TR (MOD) plan. (4) The total net revenue of the TR(MOD) plan should be increased by 8.1% as compared with the TR plan. The benefits will be much greater for analyses over longer periods. Overall, the DSR plan will allow for an improved socio-economic environment for new businesses to support disrupted workforces during and after mining.

## 1. Introduction and Problem Statement

Mineral extractive industries are an important global industry and form the foundations of our lives [1]. World Mining Data indicates that the industry extracted over 17-billion tons of raw materials with a value of about USD 2.03 trillion or about 2% of the global GDP in 2022 [2]. The industry is expected to grow consistent with societal needs [3]. During minerals-related production activities, our land [4,5], water [6,7], and air and ecosystem resources [8] are disturbed short-term and can also be negatively impacted long-term unless disturbed areas are appropriately reclaimed. Society supports mineral extraction activities since they have the advantages of significant economic, social, and political impacts. However, the industry must slowly transform under pressure from global competition, environmental regulations, and local communities to be sustainable [9].

Coal is the main energy resource [10] in China, with a production rate of about 4.5 b. tons in 2022 [11]. It has contributed to the national economic development over the last few decades [12]. Underground coal mining using longwall techniques is very common in China [13] and accounts for 90% of coal production [14]. Due to different geologic conditions, other major coal producing countries mainly use surface mining, such as the USA, India, Australia, and Russia [15]. Surface subsidence impacts caused by underground coal mining vary in different regions. For instance, surface subsidence in USA is only about 0.5 to 2 m [16], and face advance rates are about 30 m per day [14]. A small amount of grading and water ditches development is performed to use land for agriculture. In China, the reclamation of surface subsidence poses more serious problems [14]. There are several regions where underground coal mining is performed with shallow groundwater and large surface subsidence [17]. For example, the deepest mining subsidence of 19.8 m was observed in Huainan city of Anhui Province [18]. In such areas, subsidence can negatively impact on farmland [19], the landscape topography [20], surface and ground water resources [21,22], ecosystem [23], and socio-economic health both during and upon completion of mining activities. It can also lead to loss in agricultural productivity [24] and production [25,26] and threaten food security, if reclamation is not appropriately planned.

Currently, reclamation operations in China are mainly focused on relatively stable mining-subsided land after mining [27], and they are known as a traditional reclamation (TR). Therefore, a large amount of agricultural land has already submerged into water, including the fertile topsoil and subsoil. Moreover, reclamation operations in waterlogged areas can also increase the reclamation cost by about 30% [28]. Backfill materials with characteristics similar to submerged soils are not plentiful for reclaiming large subsided areas and can further increase the costs. Therefore, TR can result in the loss of farmland during the active mining period, a low percentage of reclaimed farmland in post-mining land use, poor quality of farmland, and higher reclamation costs. These deficiencies in TR led to exploring alternate reclamation technologies for actively subsiding mining areas.

Through careful planning, it is possible to successfully reclaim and restore the disturbed ecosystems [29] and close a mine with minimal or no long-term impacts. Based on the reviews of the TR plan, China University of Mining and Technology Beijing (CUMTB) researchers extended the concurrent mining and reclamation concept to subsiding lands [30] and demonstrated its advantages through a case study in Huaibei city. While mining in areas with a shallow groundwater table and agricultural resources, dynamic subsidence reclamation (DSR) [31] immediately ahead of and behind the longwall face evolved to maximize the utilization of land and water resources for agriculture. Zhao and Hu [32] proposed pre-reclamation concepts and developed a generalized technical model for its implementation. Xiao [33] analyzed reclamation time periods for actively subsiding areas to plan for concurrent mining and reclamation. Using time-based subsidence projections and reclamation operation requirements, Hu and Xiao [34] developed an optimized concurrent mining and reclamation plan for a case study in north Anhui. Xiao et al. [35] suggested that topsoil must be stripped before it submerges into subsiding waterlogged areas and should be used to create alternate farmland areas behind the face or around the edges of subsiding areas to minimize the loss of farmland. Xiao et al. [36] introduced concurrent mining and reclamation technology in eastern China for prime farmland protection. Hu et al. [29] extended DSR approaches to minimize impacts on farmland and water resources with a simple economic analysis. Zhang [37] researched the techniques for concurrent mining and reclamation planning in multiple coal seams mining areas. Chugh [14] contributed to the above research and developed a state-of-the-art review paper on concurrent mining and reclamation concepts and their applications in China. Chen and Hang [38] proposed a soil reconstruction procedure for dynamic reclamation. Li et al. [10] integrated coal mine and reclamation planning to reduce impacts of subsidence on mined lands and identified four key steps for planning and interactions among them. Through analyses of alternate mining plans, Feng et al. [39] developed optimum mining plans for the Guqiao coal mine. Li et al. [40] considered coal production and aboveground development or protection to optimize the layout of underground coal mining in Jining city of China. Previous research mainly focused on reclamation time, modifying mining planning, multiple coal seams, and the soil reconstruction procedure.

DSR planning discussed here advances previous DSR tools to mitigate both environmental and socio-economic negative impacts. In the DSR technology, reclamation unit operations (topsoil and subsoil removal and replacement, farming and crop harvesting, and management of water resources) are dynamically implemented ahead of and behind the current mining areas to minimize negative impacts to land and water resources and to nurture new and/or old socio-economic enterprises in reclaimed areas.

## 2. Materials and Methods

### 2.1. Hypothesis and Goals

Several authors [14,31] have assessed the benefits of DSR for a few mines in China. An opportunity developed to perform similar studies at another case study mine presented here. The authors extended the earlier planning concepts to include creative ideas, such as: (1) the development of super-farmland areas that should have higher agricultural productivity compared to pre-mined lands to offset the loss of some farmland in water. These areas would have thicker topsoil and subsoil replacement and better farm management practices in marginally impacted subsided areas and create business and employment opportunities for displaced workers, both in the short-term and long-term; (2) intentionally creating strategically located large water resources areas that could be used to support economic development through new towns, recreational sports, and water resources management facilities for multiple towns within the region; and (3) supporting community development efforts to positively impact socio-economic development during and after the mining ceases in the area. Items (2) and (3) above are considered very important now for planning for the closure of mines after active mining.

Since the case study area had already been mined and reclaimed using the TR plan, the goal of this paper is to assess what benefits could have been achieved if the mining and reclamation had been performed using the authors’ proposed TR-modified (TR (MOD)) approaches and DSR concepts. In the TR(MOD) concept, the soils are stripped ahead of mining and used to increase the amount of agricultural land. The research includes analyses and discussions with community leaders in the region to document future needs for improved ecosystems and socio-economic development for the community. Even though scientific economic comparisons could not be undertaken, it was thought that even simplistic subjective comparisons here should lead to meaningful DSR concept implementation projects. It should also help assess the relative importance and difficulties in implementing proposed DSR concepts.

### 2.2. Surface Description of the Case Study Area

The case study area is located in the northwest part of the Zhangji coal mine in Anhui Province, China (Figure 1). The relatively flat land represents the alluvial plains of the Huaihe River, with surface elevations varying from +17.3 m to +26.5 m above the mean sea level (MSL) (Figure 1a). Additionally, surface slopes are no more than 5°. The topsoil and subsoil thicknesses in the area average about 0.5 m and 1.0 m, respectively. The ground water level (GWL) is about 1.5 m below the ground surface. The area is in a semi-humid monsoon/warm climate zone with four different seasons, an average annual temperature of 15.1 °C, and 926 mm of rainfall occurring mostly in summer (from June to August). The average wind speed is 3.18 m/s, with southeast and east winds in spring and summer, southeast and northeast winds in fall, and northeast and northwest winds in winter.

Farmland accounts for about 69.6% of pre-mining land use for cultivating rice and wheat (Figure 1b and Table 1), with a multiple cropping index of 200%. Rice usually grows from early June to late September, while wheat is planted from October to June. The estimated production of rice and wheat are 7500 kg and 6750 kg per hectare (ha) per year, respectively. In addition, cucumber and tomato vegetables can be planted in the local area, with production rates of 15,000 and 22,500 kg, respectively, per ha per year. The Huainan government [41] has indicated the sale price for rice, wheat, cucumber, and tomato to be about 2.68 Renminbi (RMB)/kg, 3.08 RMB/kg, 9.66 RMB/kg, and 7.62 RMB/kg, respectively. The revenue for water resources was about 2.5 RMB/m^3^ in 2022.

Most people in the study area are farmers. Some businesses breed fish in small ponds. Carp, grass carp, and crucian carp are commonly produced twice a year, with total production rate of about 4500 kg/ha. Huainan Agricultural Products indicated the prices of carp, grass carp, and crucian carp to be 12, 16, and 18.5 RMB/kg, respectively, in 2022 [41]. Several people work in the case study coal mine and two other factories—the Xueyao building material factory in the northeast and the Guanyin rotary kiln factory for firing red brick in the southwest.

### 2.3. Mining Practices and Subsidence Analysis for the Case Study Area

Five single-seam longwall faces (P1–P5) were mined in the area during the period 2015–2020 (Figure 2), with an average mining thickness of 6.0 m and a seam dip of about 6°, as shown. The mining depth varied from 480 m to 575 m. The underground mining area within the boundary was about 127.9 ha. The underground mining areas for Panels 1–5 are about 28.6 ha, 27.2 ha, 26.4 ha, 29.7 ha, and 16.0 ha, respectively. The underground mining area is about 38% of the case study area (333.4 ha). The probability integration approach is used to project surface subsidence after the mining of each panel with consideration of the original terrain. The subsidence projection parameters are shown in Table 2 and Figure 3.

With mining progress, the subsidence-influenced land and waterlogged areas will increase gradually (Figure 4 and Table 3). However, farmland area will continue to decrease due to mining subsidence. The maximum projected subsidence after all mining is about 4.4 m, with the final surface area influenced by mining as 333.4 ha (Table 3), which is the case study area.

It is projected that where the post-mining surface elevation is less than +17.7 m, the land would get submerged under water due to mining subsidence. During panel 1 mining, some regions would submerge into water and land use patterns would begin to change. Mining and water-submerged land areas continue to grow with additional mining, as shown in Figure 4. After the mining of panel 5, subsidence waterlogged areas will account for 36.0%, while the farmland area will decrease from 69.6% pre-mining to 43.0% post-mining (Table 3). The mining of all five panels will result in: (1) the loss of about 26.6% of farmland for cultivation; and (2) increasing waterlogged areas to about 36.0% of the case study area (Figure 4 and Table 3). These impacts will also have negative impacts on the socio-economic health of the communities in the region, both during active mining and after mining ceases. Furthermore, erosion, sedimentation, and acid mine drainage potential are likely to develop in some areas due to coal composition. Water resources would likely have high dissolved and suspended solids, and the utilization of water resources would involve additional treatment costs as well.

### 2.4. TR, TR (MOD), and DSR Reclamation Practices

#### 2.4.1. TR Reclamation Planning

This would involve grading the land surface appropriately after mining panel 5, replacing the available soils around farmland areas, and vegetating the areas after surface movements have stabilized. In this plan the topsoil and subsoil are not removed during active mining and backfilling. The proposed reclaimed farmland areas are designed to have a surface elevation of +21.2 m MSL based on the case study area characteristics. Soils in and around shallow water-submerged areas (cut area with surface elevations higher than +17.2 m and less than +17.7 m) will be stripped down to an elevation of +17.2 m, and these soils will be backfilled into areas with relatively small subsidence to reclaim them as farmland (filling area). Soils stripped from areas with small subsidence (stripping area with surface elevation higher than +21.2 m) will then be reclaimed as farmland. Using appropriate post-subsidence topographic maps and the above design values, it is estimated that about 0.04 million cubic meters (m cu. m) and 3.05 m cu. m of soils will be obtained from the cut and stripping areas. Thus, about 3.09 m cu. m of soils will be backfilled in filling areas to reclaim them as farmland. Upon completion of all reclamation, it is projected (Figure 5) that: (1) there will be about 240.7 ha of farmland or about 72.2% of the case study area; and (2) 92.7 ha of water reservoir in the center with about 3.0 m cu. m of water in it. Upon reclamation, revenue sources will include rice and wheat planted on reclaimed farmland, water resources in the reclaimed water reservoir, and small businesses that might develop after reclamation is completed.

#### 2.4.2. TR (MOD) Reclamation Planning

In order to modify the TR plan as a TR (MOD) plan, it was planned to strip soils ahead of mining and store them before they will submerge into water. However, these soils will not be used for reclamation until year 6, after all mining is completed. This will allow agricultural production on soils until just before they are projected to be submerged in water. A brief description of plans while mining each panel is given below.

Mining panel 1: Subsidence projections show that the central red area (Figure 6, panel 1) will subside below the groundwater table. Therefore, prior to subsidence, soils in the red area (cut area C) will be stripped in advance to achieve an elevation of +17.2 m. About 0.59 m cu. m of soils will be stripped from this area to form a water reservoir. Area C will eventually form a deep-water reservoir after the mining of panel 1. The stripped soils will be stored in the brown area (B) on both ends of the proposed water reservoir. Appropriate diches will be constructed to direct the surrounding surface water into area C to protect the current farmland.

Mining panel 2: During this mining, soils in area C adjacent to the blue water reservoir will be stripped. About 0.13 m cu. m of soils are projected to be available here and will be stored in area B. The water reservoir area will be expanded toward the southwest. Some additional diches will be needed to direct the surrounding surface water into area C and to protect the current farmland.

Mining panel 3: This panel is adjacent to panel 1 in the northwest mining area. Therefore, about 0.15 m cu. m of soils in area C will be stripped ahead of mining to extend the water reservoir toward the northwest direction. The stripped soils will be stored in area B, and as before, some ditches may be needed to channel surface waters into area C and to protect the current farmland.

Mining panel 4: Waterlogged areas due to mining subsidence are not projected to increase much during this mining since panel 4 is around the center of panels 1 and 2. Only a small area C in the southeast corner is expected to get submerged. Before that, about 0.02 m cu. m of soils will be stripped and stored in area B and some ditches will channel surface waters into area C to protect current farmland.

Mining panel 5: During this panel mining, waterlogged areas will extend northwest. About 0.05 m cu. m of soils will be stripped from area C and stored in area B, and ditches will channel surface water into area C to protect current farmland.

After mining panel 5: After the mining of all five panels, reclamation activities involving the grading and replacement of soils will begin. The green areas with a relatively small amount of subsidence will be backfilled with soils stored in area B over the past five years to achieve an elevation of +21.2 m for cultivating it as farmland (filling area F). Since the purple area has higher elevation than the designed farmland elevation, it will be stripped down to a +21.2 m level (stripping area S) with about 3.05 m cu. m of subsoil obtained from this area. The stripped soils from areas B and S will be spread out over area F to reclaim it as farmland.

During the entire reclamation process, about 3.99 m cu. m of soil will be stripped and backfilled. In the TR(MOD) plan, the entire mining area will be reclaimed to 259.5 ha of farmland (77.9% of the case study area), and 73.9 ha of water reservoir will form around the center of the mining area, with a volume capacity of 3.85 m cu. m of water. This amounts to be about 5.6% more farmland available than in the TR plan. The water resources will however be reduced from 92.7 ha of water reservoir in the center in the TR plan to 73.9 ha in the TR(MOD) plan. As in the TR plan, revenue sources will include rice and wheat on reclaimed farmland, water resources in the reclaimed water reservoir, and any small businesses that may develop after reclamation.

#### 2.4.3. DSR Reclamation Planning: Concepts and Implementation

Here, reclamation unit operations (topsoil, subsoil removal and replacement, farming and crop harvesting, and harnessing water resources) are dynamically implemented concurrent with mining ahead of and behind the projected subsidence areas. This is done to minimize land and water resource impacts and to nurture the construction of new and/or old socio-economic enterprises as mining progresses. Since topsoil and subsoil are relatively rich in organic matter and are critical for plant growth, these are stripped separately before the land submerges into water during the mining of each panel (Figure 7). The soils are backfilled in planned farming areas concurrent with mining. In planning DSR, two novel concepts are also considered by authors.

(1)The development of super-farmland of high agricultural productivity to offset the potential loss of land area for farming and productivity in farming areas. In these areas, the separately stripped topsoil and subsoil will be spread out to have much larger thickness of soils than typically found on farmland. These areas will be further augmented by productivity-enhancement agricultural practices. The desired result will be to restore and/or to increase agricultural production to a higher quality and volume from DSR practicing on small reclaimed areas. These areas may also be used for planting vegetables and fruits for distribution in the region through cooperative Farmers Markets programs and for developing small businesses to enhance socio-economic development activities.(2)The development of planned water accumulation reservoir/s to serve regional water supplies for different uses such as drinking water, irrigation, recreational sports, fish hatcheries, and other needed small businesses. These will be developed in concert with regional community leaders to support socio-economic planning post mining. DSR plans are illustrated below using Figure 7.

Mining panel 1: Subsidence projections show that the central red area will subside below the groundwater table (Figure 7, panel 1). Therefore, prior to subsidence, soils in the red area (cut area C) will be stripped in advance to achieve an elevation of +17.2 m. About 0.22 m cu. m of topsoil and 0.37 m cu. m of subsoil will be stripped from area C to form a water reservoir. Concurrently, the green area on both ends of the proposed water reservoir will be backfilled to reclaim it as farmland (filling area F). The subsoil from area C will be moved to backfill area F first. Then, the stripped topsoil from area C will be used to backfill area F to develop a highly productive super-farmland (SR) which will have about two times the typical topsoil thickness found on farmland. Area C will form a deep-water reservoir (reclaimed water area W) after the mining of panel 1.

Mining panel 2: During this panel mining, soils in area C adjacent to the blue water reservoir W will be stripped. About 0.06 m cu. m of topsoil and 0.07 m cu. m of subsoil should be available here. The water reservoir area will expand southwest. Simultaneously, area F adjacent to the area W will be backfilled to reclaim it as a super-farmland. Thus, the stripped subsoil from area C will be spread out in area F first. Then, the stripped topsoil from area C will be used to backfill area F to create as SR with about two times the typical topsoil thickness on farmland.

Mining panel 3: This panel is adjacent to panel 1 in the northwest mining area. Therefore, soils in area C will be stripped ahead of mining. About 0.07 m cu. m of topsoil and 0.08 m cu. m of subsoil will be stripped to expand the water reservoir toward a northwest direction. Simultaneously, area F adjacent to area C will be backfilled to reclaim it as a super-farmland. The stripped subsoil from area C will be backfilled in area F first. Then, the stripped topsoil from area C will be spread out over area F to create a super-farmland with about three-times the usual topsoil thickness.

Mining panel 4: During this mining, waterlogged areas due to mining subsidence are not projected to increase much, since panel 4 is around the center of panel 1 and 2. Only a small area C in the southeast corner is expected to get submerged in water. Before that occurs, about 0.01 m cu. m topsoil and 0.01 m cu. m of subsoil will be stripped separately. Simultaneously, area F adjacent to it will be backfilled with the above soils to reclaim it as a super-farmland. The stripped subsoil from area C will be spread out over area F first. Then, the stripped topsoil from area C will be spread out over area F to create a super-farmland with about 1.1 times the usual topsoil thickness.

Mining panel 5: During this mining, waterlogged areas will spread toward the northwest direction. Therefore, in area C 0.02 m cu. m of topsoil and 0.03 m cu. m of subsoil will be stripped separately. Simultaneously, area F will be backfilled with the above soils to achieve an elevation of +21.2 m for farmland production. Since the purple area has higher elevation than the designed farmland elevation, it will be stripped down to +21.2 m. The purple area is named as stripping area S. To achieve this, about 0.56 and 0.59 m cu. m of topsoil will be stripped first from areas F and S, respectively. About 3.05 m cu. m of subsoil will then be obtained from area S. Then, stripped subsoils from areas C and S will be spread-out over area F. Finally, the stripped topsoil from area C and F will be backfilled over area F to reclaim it as usual farmland. Stripped topsoil from area S will also be backfilled into area S to reclaim it as usual farmland.

Using the above DSR approaches, the entire mining area will be reclaimed to 259.5 ha of farmland (77.9% of the case study area), and 73.9 ha of water reservoir will form around the center, with a volume capacity of 3.8 m cu. m of water. In the entire reclamation process, about 4.34 m cu. m of topsoil and subsoil will be stripped and backfilled. To support local socio-economic development and create more job opportunities, cucumber and tomato vegetables will be planted on reclaimed super-farmland, and rice and wheat on reclaimed usual farmland. Simultaneously, based on input from the local community organizations, other businesses can be developed to breed fish or develop water sports around the developed water reservoir.

## 3. Results

### 3.1. Land Resources

In the DSR plan, the total reclaimed farmland area is always larger during the entire mining period than in TR and TR(MOD) plans since reclamation operations are conducted concurrently ahead of and behind the mining face and adjoining areas. Such increases persist until year 5, after which the differences between DSR and TR/TR(MOD) plans start to decrease because reclamation is initiated in the TR/TR(MOD) plans in year 6 (Table 4). Upon completion of all land reclamation in the three plans, the farmland area is still 5.6% larger in DSR than in the TR plan. However, in TR(MOD), because of stripping soils before they submerge into water, the farmland area is the same as in the DSR plan. It is noted that stripping soils before they submerge into water is very important for farmland reclamation and long-term development.

### 3.2. Water and Fishery Resources

In the DSR and TR(MOD) plan, waterlogged areas due to mining subsidence are smaller than that in the TR plan during the five-year mining period, since soils are being stripped to form a deeper water reservoir. However, the water resource volume is much larger than in the TR plan because of the deeper depth of the water in the waterlogged areas. In the DSR plan, since stripped soils are backfilled concurrently with mining to create farmland, the water resource volume is larger than in the TR(MOD) plan until year 5. Starting in year 6, soil backfilling is performed in the TR(MOD) plan, and water resource volume is the same as in the DSR plan. Upon the completion of reclamation in the three plans, there is about 0.8 m cu. m more of water resources in the DSR and TR(MOD) plans, or about 30.2% more water than in the TR plan (Table 5).

In the DSR plan, businesses will be developed to breed fish and introduce water sports in the developed water reservoir, which were not designed in previous DSR research [31]. Based on the local situation, the carp, grass carp, and crucian carp can be mixed bred in the ratio of 3:1:1. The expected three types of fish productions from the reservoir are shown in Table 6. An estimate of the revenues from fish production is included later.

### 3.3. Agricultural Resources

In this discussion of agriculture production in the TR, TR(MOD), and DSR plans, it is assumed that only rice and wheat are planted on reclaimed farmland. In the DSR plan, topsoil and subsoil are stripped and backfilled separately before they submerge into subsided waterlogged areas, which were not separated in previous research [10,13,14,30,31,34]. Therefore, the productivity of reclaimed farmland should recover quickly after reclamation and approach about 100% of the pre-mining values during the first year and beyond. In the TR(MOD) plan, such productivity will recover slower since topsoil and subsoil are mixed during the reclamation process, with projected productivity values of about 60%, 70%, 80%, 90%, and 100% over the five-year period after reclamation is initiated in year 6. In the TR plan, the productivity for the reclaimed farmland will be expected to recover even more slowly since topsoil and subsoil were mixed and some soils submerge into water during the process. The productivity of reclaimed farmland is projected to be 50%, 60%, 70%, 80%, 90%, and 100% over the six-year period after reclamation is initiated in year 6. Therefore, the differences in rice and wheat production between the DSR and TR/TR(MOD) plans will continue to increase until year 6 (Table 7), but then should decrease slowly over the following five years because of reclamation in the TR/TR(MOD) plan and higher agriculture productivity. However, due to the separating of topsoil and subsoil and more reclaimed farmland in the DSR plan, rice and wheat production is always larger than in the TR and TR(MOD) plans. Thus, separating topsoil and subsoil can significantly affect farm productivity.

In the DSR plan, cucumber and tomato are grown in super-farmland areas, which were never considered in previous research [10,13,14,28,30,31,32,33,34,35,36,37,38,39,40]. In addition, rice and wheat are planted on the reclaimed, usual farmland. Because super-farmland is created along with mining during the first 4 years, the production of cucumber and tomato crops will steadily increase (Table 8) and then be level after that. However, rice and wheat production will decrease in the first four 4 years due to the decreased amount of farmland after mining, but then increase substantially after reclamation initiation in year 5, being level thereafter.

### 3.4. Socio-Economic Impacts

Costs are primarily related to soil handling, grading for reclamation, and the fact that stripping and filling one cubic meter of soil should cost 20 RMB. In the TR plan, since reclamation is initiated in year 6, there is no reclamation cost during the first five years. However, in the TR (MOD) plan, soils will be stripped during years 1–5 and then be used to reclaim farmland in year 6; some reclamation cost will occur during years 1–6. Similarly, in the DSR plan, since reclamation costs are primarily associated with soil stripping and backfilling, all the reclamation costs will be incurred by the end of year 5. In the TR and TR (MOD) plans, benefits are accrued from farming areas of rice and wheat production and water resources in the reservoir. However, in the DSR plan, revenues come not only from rice and wheat production from reclaimed usual farmland, but also from cucumber and tomato production on super-farmland. In addition, revenues are also obtained from fish breeding associated with the water reservoir. The estimates of the costs and benefits are shown in Table 9. These costs do not account for value of money over time and price inflation over later years and are therefore conservative and simple costs; these data therefore favor TR reclamation in comparison. Over the 11-year period, the total net revenue in the DSR plan is 2.8 times and 1.2 times more than that in the TR and TR (MOD) plans, respectively. The total net revenue of the TR(MOD) plan should increase by 8.1% as compared with the TR plan. The benefits will be much greater for analyses over longer periods. The net revenue in the DSR plan is always larger than in the TR and TR(MOD) plans except for year 5, since a lot of reclamation and soil handling activities are completed during this year. Similarly, the difference in net revenue between DSR and TR/TR(MOD) is greatest in year 6, when most reclamation work is initiated in the TR and TR(MOD) plans. Thus, one would expect much larger economic benefits with the DSR plan when analyzing performance over a longer period of time, which was not considered in previous research [10,13,30,32,33,34,35,36,37,38,39,40].

## 4. Discussion

With significant expected coal production from underground mining in China over the next few decades and global pressures for sustainable development, there has been significant advances in research on reclaiming unstable subsided lands. Underground mining environments similar to China are not common globally and therefore most research on the topic has been in China, and it has been slow and evolutionary. As discussed earlier, Zhao [32] and Xiao [33] analyzed the initiation of reclamation time without considering the economic benefits. Zhang [37] extended the concepts to mining multiple coal seams. Li et al. [10,13] and Feng et al. [39] focused on underground mine plans to minimize subsidence impacts without considering original surface topography, and Chen and Hang [38] focused on soil reconstruction procedures. Hu et al. [31] developed DSR concepts for mining areas with shallow groundwater resources without considering topsoil and subsoil separation, socio-economic development concepts of the super-farmland, large water storage, and associated small- and large-scale business development. So, the authors here have significantly advanced DSR planning concepts to include: (1) creating super-farmland areas of high agricultural productivity to offset the loss of farmland submerged under water, and planting fruits and vegetables to diversify and enhance revenue streams; (2) creating large areas of planned quality water resources to serve existing and new or improved socio-economic ventures to support long-term community needs at the local and regional levels, and creating new water sports businesses and fish breeding areas; (3) perform simple cost–benefit analyses for modified DSR concepts; (4) develop a TR(MOD) plan to analyze the importance of stripping soils before they submerge into water and separating topsoil and subsoil; and (5) consider pre-mining surface topography to predict subsidence and develop the DSR, TR, and TR (MOD) reclamation plans based on dynamic post-mining topography. To the best of the authors’ knowledge, such research is not being conducted elsewhere in the world, since such mining conditions do not exist. An invited oral presentation was made by Dr. Chugh to World Mining Congress professionals in India in 2019 (no publication) and received significant interest.

Previous limited DSR concept field implementations in China [30,31,34] led to the following observations: (1) the soundness of the technical concepts was well received; (2) the farmers were hesitant to strip topsoil and subsoil prior to the soil being submerged in water due to reduced agricultural production; (3) mining companies allocated more resources to production than to DSR planning with less than optimum efficiencies in the reclamation processes; and (4) socio-economic considerations and community involvement were not considered. The above was to be expected considering the newness of the concepts.

The authors propose to widely disseminate these research findings to mining professionals and to government agencies responsible for regional development to identify project opportunities for implementing these concepts in single seam and multiple-seam mining areas. Such projects should also consider optimizing both mine and DSR planning to minimize the production cost to the consumer. The authors hope that this study will encourage mine operators to implement DSR concepts in planning while considering new and ongoing mineral development projects. DSR is a powerful engineering and planning concept that can minimize the land, water [14,31], and air impacts of mining and enhance socio-economic conditions under a variety of mining conditions. Furthermore, it should not be limited to only reclamation planning. Mining and DSR planning should be integrated to develop both optimal mining and reclamation plans [10,13,39] to enhance the profitability of the mining venture and its sustainability. That should further develop new ideas for both mining and reclamation for the improved sustainability of mineral projects. Toward this goal the authors recommend that every mining project, irrespective of the stage of its development, should consider having a steering committee consisting of mining, business, and regional development professionals to review alternate opportunities for mining and reclamation and to improve project sustainability while maximizing the profit potential. That would also ensure the development of sound mine closure plans [42] and the slow implementation of the idea that “mine closure must begin the day mining starts” while maximizing profit potential.

## 5. Conclusions

This paper has attempted a conceptual implementation of DSR planning concepts at a large coal mine involving five single seam longwall faces in a mining area with a shallow groundwater table about 1.5 m below the ground surface and 4.4 m of maximum surface subsidence due to mining. The study involved mining and post-mining reclamation to assess if DSR could have improved both the environment and socio-economic conditions for post-mining land use as compared to using TR approaches in China. In DSR topsoil, subsoil, farming, and water resources management were dynamically synergized concurrently with mining. Within TR, two approaches were considered: (1) designated as TR, all reclamation activities are initiated only after all mining had been completed in the area, and land was allowed to submerge into water in the subsiding areas; and (2) TR (MOD), where soils (topsoil and subsoil) ahead of mining, likely to be submerged, were stripped and stored without separation into topsoil and subsoil for reclamation after all mining was completed in the area. The authors undertook the current analyses for this hypothetical case as soon as the data became available.

A comparison of the three analyzed plans shows that: (1) upon final reclamation, farmland area and water resources in the DSR and TR (MOD) plans are increased by 5.6% and 30.2%, respectively, as compared to the TR plan. Stripping soils before they submerge into water is important for farmland reclamation and long-term economic development; (2) due to topsoil and subsoil separation in DSR plan, reclaimed farmland productivity should recover quickly and agriculture production would be larger than in the TR and TR(MOD) plans; 3) the total estimated net revenue in the DSR plan should be 2.8 times more than in the TR and 1.2 times more than that in the TR (MOD) plan; and (4) the total net revenue of the TR(MOD) plan should be increase by 8.1% as compared with the TR plan. The above net revenue benefits should be even greater for analyses over longer periods. Furthermore, in DSR, the availability of farmland and water resources will be for much longer throughout the active mining period. The mining and reclaimed areas in DSR can therefore provide an improved socio-economic environment through community business development and the settlement of labor markets both during mining and after mining ceases in the area. Although solutions and sample calculations presented are simple, the paper clearly demonstrates the usefulness of the DSR concepts to minimize negative impacts to the environment while enhancing the long-term socio-economic environment.

## Figures and Tables

**Figure 1 ijerph-20-05213-f001:**
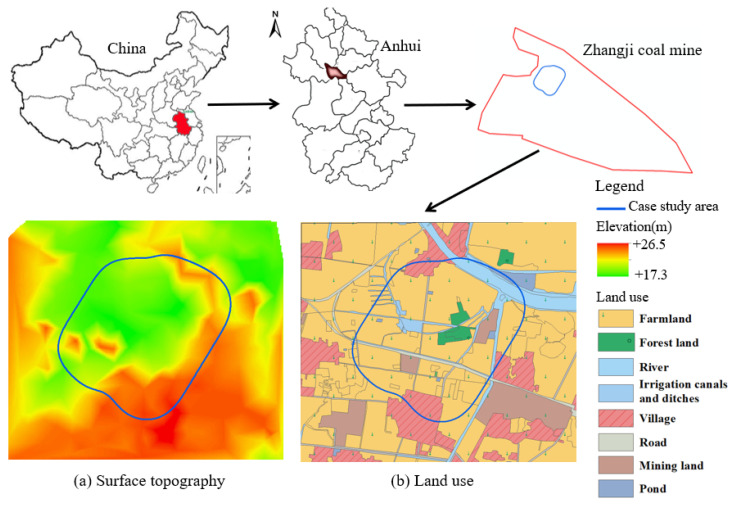
Location, surface topography, and land use of case study area.

**Figure 2 ijerph-20-05213-f002:**
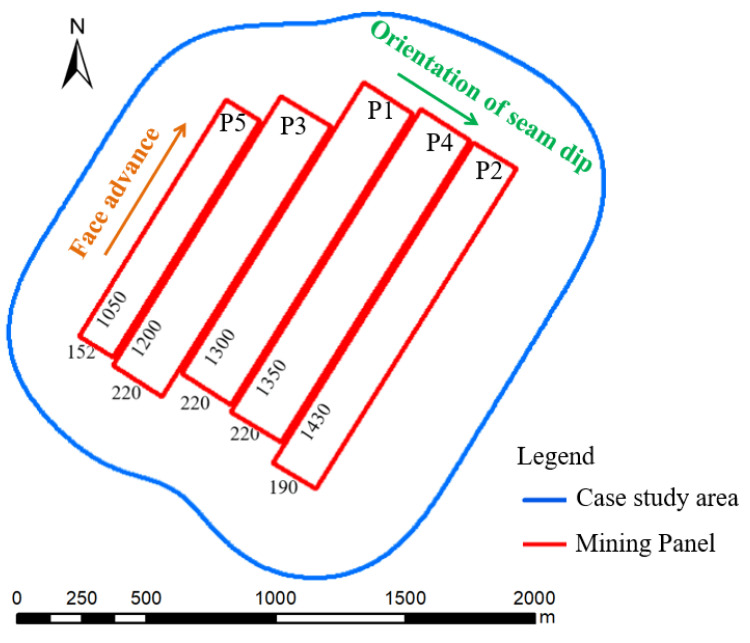
Location of panels and mining area boundaries.

**Figure 3 ijerph-20-05213-f003:**
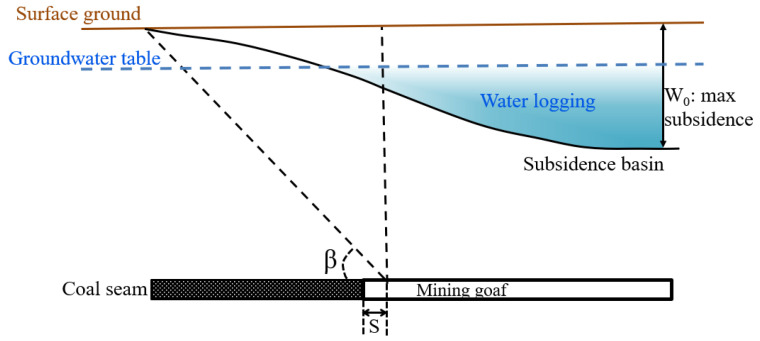
Surface subsidence diagram showing characteristic variables.

**Figure 4 ijerph-20-05213-f004:**
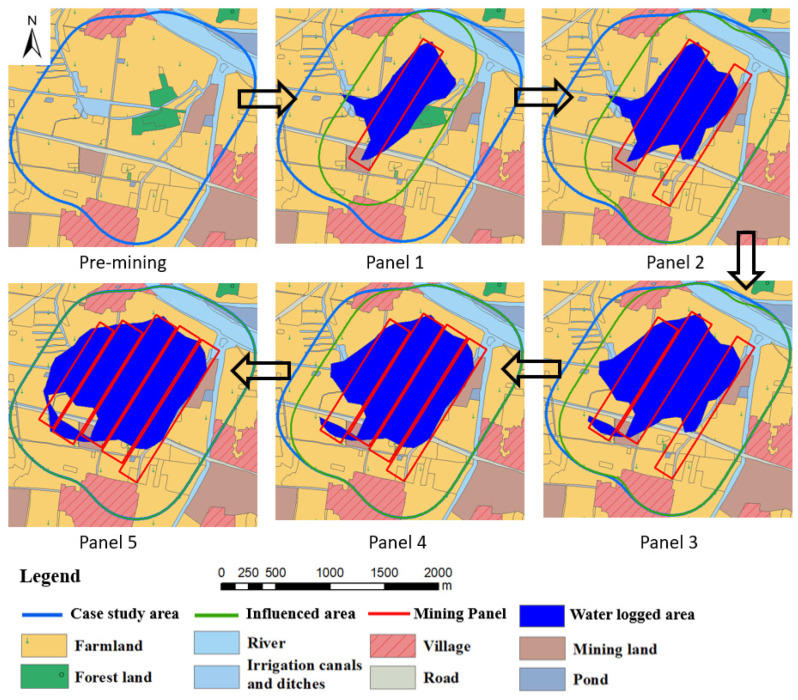
Subsidence influence on land use after mining of each panel.

**Figure 5 ijerph-20-05213-f005:**
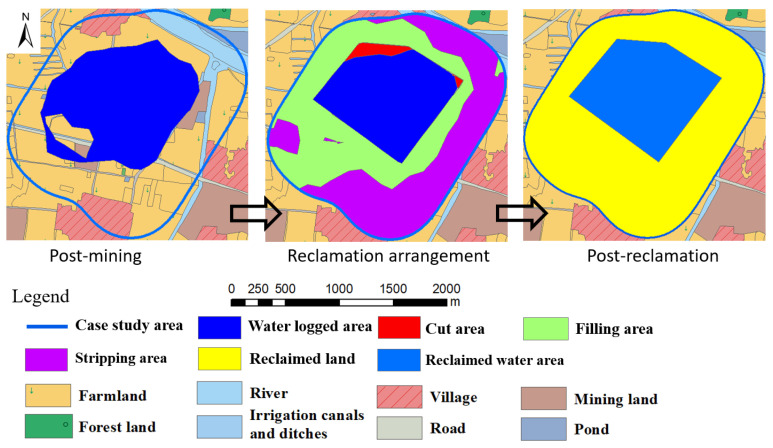
Reclamation plans for TR plan.

**Figure 6 ijerph-20-05213-f006:**
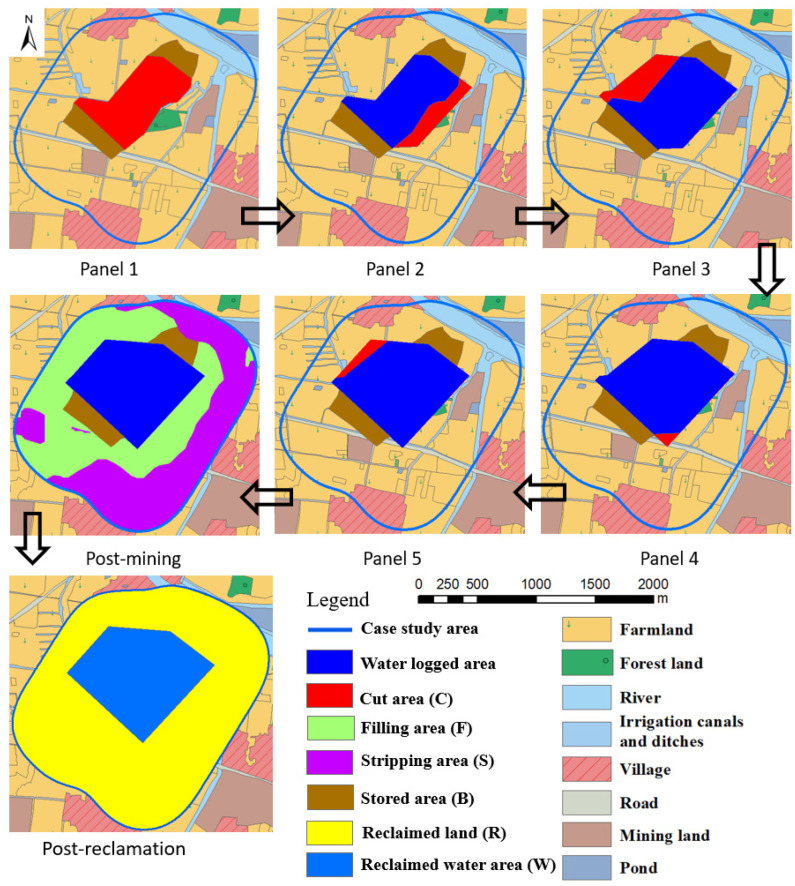
Reclamation planning of TR(MOD) after each mining panel.

**Figure 7 ijerph-20-05213-f007:**
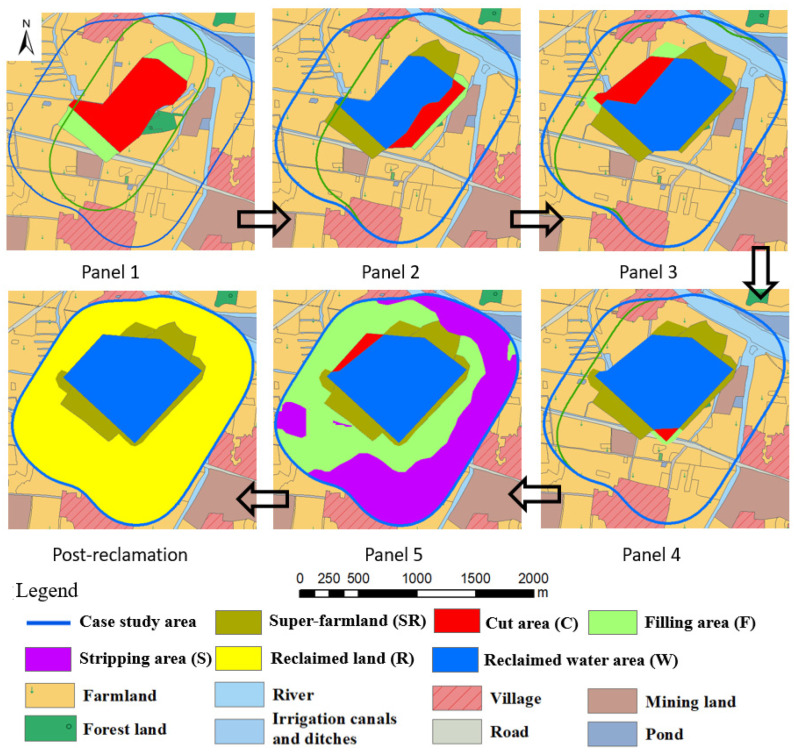
DSR planning for each mining panel.

**Table 1 ijerph-20-05213-t001:** Pre-mining land use for the projected subsidence-influenced areas.

Land Use Pattern	Area (ha)	Proportion (%)
Farmland	231.9	69.6
Forest land	11.6	3.5
River	24.9	7.5
Irrigation canals and ditches	16.1	4.8
Village	22.4	6.7
Road	11.2	3.4
Mining land	12.1	3.6
Pond	3.0	0.9
Total	333.4	100.0

**Table 2 ijerph-20-05213-t002:** Important subsidence projection parameters.

Parameters	Value
Subsidence factor of mining: q = W_0_/mcosα *	0.78
Tangent of main influence angle: tanβ	1.7
Horizontal displacement factor: b	0.3
The displacement distance: S (m)	10
Influence propagation angle: θ (deg.) = 90 − 0.6α	86.4

* W_0_: max subsidence, α: coal seam dip.

**Table 3 ijerph-20-05213-t003:** Distribution of waterlogged areas with mining progress.

Mining Panel	Max Subsidence (m)	Influenced Area (ha)	Farmland		Waterlogged Area	
			Area (ha)	Proportion (%)	Area (ha)	Proportion (%)
1	2.9	158.4	199.8	59.9	46.2	29.2
2	3.0	260.5	188.2	56.5	66.2	25.4
3	4.0	305.8	168.1	50.4	90.2	29.5
4	4.4	308.4	158.5	47.6	103.2	33.4
5	4.4	333.4	143.4	43.0	120.1	36.0

**Table 4 ijerph-20-05213-t004:** Farmland land resource in three reclamation plans.

Year	TR Plan		TR(MOD) Plan		DSR Plan			
	Area (ha)	Proportion (%)	Area (ha)	Proportion (%)	Super-Farmland (ha)	Usual Farmland (ha)	Total Area	Proportion (%)
1	199.8	59.9	186.5	56.0	17.7	186.5	204.3	61.3
2	188.2	56.5	180.1	54.0	22.3	177.8	200.1	60.0
3	168.1	50.4	165.8	49.7	27.0	161.1	188.1	56.4
4	158.5	47.6	164.5	49.4	28.4	159.2	187.6	56.3
5	143.4	43.0	162.0	48.6	28.4	231.0	259.5	77.8
6	240.7	72.2	259.5	77.8	28.4	231.0	259.5	77.8

**Table 5 ijerph-20-05213-t005:** Water resources in three reclamation plans.

Year	TR Plan			TR(MOD) Plan			DSR Plan		
	Area (ha)	Volume (m. cu. m)	Average Water Depth (m)	Area (ha)	Volume (m. cu. m)	Average Water Depth (m)	Area (ha)	Volume (m. cu. m)	Average Water Depth (m)
1	46.2	0.6	1.3	43.2	0.9	1.9	43.2	1.2	2.6
2	66.2	0.8	1.7	54.2	1.1	2.4	54.2	1.5	3.3
3	90.2	1.4	3.1	68.2	1.8	3.9	68.2	2.3	5.0
4	103.2	2.5	5.4	69.7	2.9	6.2	69.7	3.4	7.3
5	120.1	2.9	6.3	73.9	3.4	7.5	73.9	3.8	8.3
6	92.7	3.0	6.4	73.9	3.8	8.3	73.9	3.8	8.3

**Table 6 ijerph-20-05213-t006:** Projected fish production in DSR plan.

Year	DSR Plan			
	Area (ha)	Carp (m. kg *)	Grass Carp (m. kg)	Crucian Carp (m. kg)
1	43.2	0.233	0.155	0.155
2	54.2	0.293	0.195	0.195
3	68.2	0.368	0.245	0.245
4	69.7	0.376	0.251	0.251
5	73.9	0.399	0.266	0.266
6	73.9	0.399	0.266	0.266

* m. kg: million kg.

**Table 7 ijerph-20-05213-t007:** Agriculture production in three reclamation plans.

Year	TR Plan		TR(MOD) Plan		DSR Plan	
	Rice	Wheat	Rice	Wheat	Rice	Wheat
1	1.50	1.35	1.40	1.26	1.53	1.38
2	1.41	1.27	1.35	1.22	1.50	1.35
3	1.26	1.13	1.24	1.12	1.41	1.27
4	1.19	1.07	1.23	1.11	1.41	1.27
5	1.08	0.97	1.22	1.09	1.95	1.75
6	0.90	0.81	1.17	1.05	1.95	1.75
7	1.08	0.97	1.36	1.23	1.95	1.75
8	1.26	1.14	1.56	1.40	1.95	1.75
9	1.44	1.30	1.75	1.58	1.95	1.75
10	1.62	1.46	1.95	1.75	1.95	1.75
11	1.81	1.62	1.95	1.75	1.95	1.75

Production unit: m. kg.

**Table 8 ijerph-20-05213-t008:** Agriculture production in DSR plans.

Year	DSR Plan			
	Rice	Wheat	Cucumber	Tomato
1	1.40	1.26	0.27	0.40
2	1.33	1.20	0.33	0.50
3	1.21	1.09	0.41	0.61
4	1.19	1.07	0.43	0.64
5	1.73	1.56	0.43	0.64
6	1.73	1.56	0.43	0.64
7	1.73	1.56	0.43	0.64
8	1.73	1.56	0.43	0.64
9	1.73	1.56	0.43	0.64
10	1.73	1.56	0.43	0.64
11	1.73	1.56	0.43	0.64

Production unit: m. kg.

**Table 9 ijerph-20-05213-t009:** Net revenue in three plans.

Year	TR Plan				TR(MOD) Plan				DSR Plan			
	Agriculture Revenue	Water Revenue	Reclamation Cost	Net Revenue	Agriculture Revenue	Water Revenue	Reclamation Cost	Net Revenue	Agriculture Revenue	Fish Revenue	Reclamation Cost	Net Revenue
1	8.17	1.53	0.00	9.22	7.63	2.24	6.84	2.58	13.23	8.16	11.80	9.59
2	7.70	1.99	0.00	9.24	7.36	2.77	2.60	7.11	14.31	10.25	2.60	21.97
3	6.87	3.62	0.00	10.10	6.78	4.52	3.00	7.90	15.14	12.88	3.00	25.02
4	6.48	6.19	0.00	12.30	6.73	7.14	0.40	13.07	15.51	13.17	0.40	28.28
5	5.86	7.28	0.00	12.80	6.63	8.62	1.00	13.86	18.45	13.97	84.60	−52.19
6	4.92	7.39	62.04	−50.02	6.37	9.62	68.00	−52.38	18.45	13.97	0.00	32.41
7	5.91	7.39	0.00	12.95	7.43	9.62	0.00	16.62	18.45	13.97	0.00	32.41
8	6.89	7.39	0.00	13.88	8.49	9.62	0.00	17.62	18.45	13.97	0.00	32.41
9	7.87	7.39	0.00	14.80	9.55	9.62	0.00	18.61	18.45	13.97	0.00	32.41
10	8.86	7.39	0.00	15.73	10.61	9.62	0.00	19.61	18.45	13.97	0.00	32.41
11	9.84	7.39	0.00	16.66	10.61	9.62	0.00	19.61	18.45	13.97	0.00	32.41
Total	79.38	64.92	62.04	82.26	88.17	83.00	81.84	89.33	187.31	142.22	102.40	227.13

Unit: million RMB (m. RMB).

## Data Availability

The data used to support the findings of this study are available from the corresponding author.

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
