# Peer review of "Sustainable Development of Underground Coal Resources in Shallow Groundwater Areas for Environment and Socio-Economic Considerations: A Case Study of Zhangji Coal Mine in China"

_ijerph, 2023, doi:10.3390/ijerph20065213_

Round 1

Reviewer 1 Report

Sustainable Development of Underground Coal Resources in Shallow-Groundwater Areas for Environment and Socio-Economic Considerations: A Case Study in China

The impact of underground coal mining on land use and agricultural system is widely distributed across mining zones. The mitigation and land reclamation measure is quiet important. This paper studied Dynamic Subsidence Reclamation in shallow-groundwater areas caused by underground coal and analyzed its advantages though contrast with two traditional reclamation plans. The topic is interesting and within the scope of the journal, the research is comprehensive. But there are still some technical issues needed be clarified before it could be accepted for publication.

1.        The authors have not clarified how this study improves existing research gaps in the introduction and abstract. Previous researches need to be systematically reviewed.

2.        Line 37-39:World Mining Data indicates that the industry extracted over 17-billion 37 tons of raw materials with a value of about $ 2.03 trillion or about 2% of the global GDP 38 in 2022.” Where does the data come from?

3.        Line 117: what TR(MOD) means?

4.        Line 160: “The underground mining area is about 38% of the total subsidence influenced (333.4 ha) area.” Here, is “the total subsidence influenced (333.4 ha) area” the case study area?

5.        Line 175 and 178: Do “Panel” and “panel” have the same meaning?

6.        Line 188: Double check the format of Table 3

7.        Line 214: For TR(MOD), when stripped soil in cut area, how much the elevation would be?

8.        Line 217 and 348: Do “Year 6” and “year 6” have the same meaning?

9.        Line 331: Figure 7 needs put here to help reader understand DSR measure.

10.    Line 432-445: The discussion part needs to be improved. What are the practical significance and impact of applicating DSR? What are the suggestions for government and coal industry?

Author Response

Comments 1: The authors have not clarified how this study improves existing research gaps in the introduction and abstract. Previous researches need to be systematically reviewed.

Responses and a summary of revisions:

Thanks for your suggestion. Introduction has been modified as follows through a systematic review of previous research to illustrate the difference between this study and previous research. See Line 75-114.

Through careful planning, it is possible to successfully reclaim and restore the disturbed ecosystems [29] and close a mine with minimal or no long-term impacts. Based on review of TR plans, CUMTB researchers extended the concurrent mining and reclamation concept to subsiding lands [30], and demonstrated its advantages through a case study in Huaibei city. While mining in areas with shallow ground water table and agricultural resources, dynamic subsidence reclamation (DSR) [31] immediately ahead of and behind the longwall face evolved to maximize utilization of land and water resources for agriculture. Zhao and Hu [32] proposed pre-reclamation concepts and developed a generalized technical model for its implementation. Xiao [33] analyzed reclamation time periods for actively subsiding areas to plan for concurrent mining and reclamation. Using time-based subsidence projections and reclamation operations requirements, Hu and Xiao [34] developed an optimized concurrent mining and reclamation plan for a case study in north Anhui. Xiao et al. [35] suggested that topsoil must be stripped before it submerges into subsiding water-logged areas and should be used to create alternate farmland areas behind the face or around the edges of subsiding areas to minimize loss of farmland. Xiao et al. [36] introduced concurrent mining and reclamation technology in eastern China for prime farmland protection. Hu et al. [29] extended DSR approaches to minimize impacts on farmland and water resources with simple economic analysis. Zhang [37] researched the techniques for concurrent mining and reclamation planning in multiple coal seams mining areas. Chugh [14] developed a state-of-the-art review paper on concurrent mining and reclamation concepts and their applications in China. Chen and Hang [38] proposed soil reconstruction procedure for dynamic reclamation. Li et al [10] integrated coal mine and reclamation planning to reduce impacts of subsidence on mined lands and identified four key steps for planning and interactions among them. Through analyses of alternate mining plans, Feng et al [39] developed optimum mining plans for Guqiao coal mine. Li et al [40] comprehensively considered coal production and aboveground development or protection to optimize the layout of underground coal mining in Jining city of China. It is clear from this review that previous research focused on time for reclamation, modified mining plans, multiple coal seam mining, and soil reconstruction procedures.

Comments 2: Line 37-39: “World Mining Data indicates that the industry extracted over 17-billion 37 tons of raw materials with a value of about $ 2.03 trillion or about 2% of the global GDP 38 in 2022.” Where does the data come from?

Responses and a summary of revisions:

Thank you for asking. It comes from “Word Mining Data 2022” which is available in the website of https://www.world-mining-data.info/. It also be added in Line 41 and Reference part of Line 578.

Comments 3: Line 117: what TR(MOD) means?

Responses and a summary of revisions:

Thanks for your suggestion. TR(MOD) is the abbreviation for TR modified reclamation approach, and it is included in Line 140-142. In this approach, soils are removed prior to their getting submerged in water in the subsiding area and are used to develop land around the subsidence basin fringes. TR(MOD) is a modification of TR where soils are not removed and are allowed to be submerged into water.

Comments 4: Line 160: “The underground mining area is about 38% of the total subsidence influenced (333.4 ha) area.” Here, is “the total subsidence influenced (333.4 ha) area” the case study area?

Responses and a summary of revisions:

Yes, it is. It has been changed into “the case study area” in Line 185. In fact, the case study area is determined based on the total subsidence influenced area. It is explained in Line 197-198.

Comments 5: Line 175 and 178: Do “Panel” and “panel” have the same meaning?

Responses and a summary of revisions:

Yes, thank you for asking. The “panel” in Line 205 has been changed to “Panel” and throughout the paper.

Comments 6: Line 188: Double check the format of Table 3

Responses and a summary of revisions:

Thanks for your suggestion. The internal border Line of Table 3 has been removed in Line 215.

Comments 7: Line 214: For TR(MOD), when stripped soil in cut area, how much the elevation would be?

Responses and a summary of revisions:

A good question. Soils in cut area C will be stripped in advance to achieve elevation of +17.2 m. It has been included in Line 246-247.

Comments 8: Line 217 and 348: Do “Year 6” and “year 6” have the same meaning?

Responses and a summary of revisions:

Yes, they do. The “Year” in Line 242 has been changed to “year”.

Comments 9: Line 331: Figure 7 needs put here to help reader understand DSR measure.

Responses and a summary of revisions:

We agree. Figure 7 has been put in Line 357 as suggested.

Comments 10: Line 432-445: The discussion part needs to be improved. What are the practical significance and impact of applicating DSR? What are the suggestions for government and coal industry?

Responses and a summary of revisions:

Thanks for your suggestion and a very good comment. Discussion and Concluding Remarks part has been rewritten and is included in the significance of DSR and suggestions for government and coal industry. See Line 463-559.

With significant continuing coal production from underground mining in China and global pressures for sustainable development, there has been a great need for increased research on reclaiming unstable subsided lands. Zhao [32] and Xiao [33] researched optimizing reclamation time but did not quantify the benefits economically. Zhang [37] advanced the above for mining multiple coal seams. Li et al. [10,13] and Feng et al. [39] worked to modify underground mine plans to minimize subsidence impacts but did not consider original surface topography in analysis. Chen and Hang [38] focused on soil reconstruction procedures. Hu et al. [31] developed DSR concepts for mining areas with shallow groundwater resources and incorporated economic analyses. However, they did not consider separating topsoil and subsoil, socio-economic development concepts of super-farm, large water storage, and associated small- and large-scale business development that are included in analyses here. So, the authors have incorporated several new concepts into DSR planning: 1) Creating super-farmland areas of high agricultural productivity to offset the loss of farmland getting submerged under water, and planting fruits and vegetables to enhance revenue streams; 2) Creating large areas of planned quality water resources to serve existing and new or improved socio-economic ventures to support long-term community needs at the local and regional levels, and creating new businesses of water sports and fish breeding areas; 3) Perform simple cost-benefit analyses; 4) In addition to TR develop a TR(MOD) plan to analyze the importance of stripping soils before they submerge into water and separating topsoil and subsoil; 5) Consider original surface topography to predict surface subsidence and develop DSR, TR and TR (MOD) reclamation plans based on dynamic subsidence topography.

This paper has attempted a conceptual implementation of DSR planning concepts at a large coal mine involving five single seam longwall faces in a mining area with shallow groundwater table about 1.5 m below the ground surface and 4.4 m of maximum surface subsidence due to mining. The study involved mining and post-mining reclamation to assess if DSR could have improved both the environment and socio-economic conditions for post-mining land use as compared to using TR approaches in China. In DSR topsoil, subsoil, farming, and water resources management were dynamically synergized concurrent with mining. Within TR, two approaches were considered: 1) Designated as TR, all reclamation activities are initiated only after all mining had been completed in the area and land was allowed to submerge into water in the subsiding areas; and 2) TR (MOD) where soils (topsoil and subsoil) ahead of mining, likely to be submerged, were stripped and stored without separation into topsoil and subsoil for reclamation after all mining was completed in the area. The authors undertook the current analyses for this hypothetical case as soon as the data became available.

A comparison of the three analyzed plans shows that: 1) Upon final reclamation, farmland area and water resources in the DSR and TR (MOD) plans are increased by 5.6% and 30.2% as compared to the TR plan. Stripping soils before they submerge into water is important for farmland reclamation and long-term economic development; 2) Due to topsoil and subsoil separation in DSR plan, reclaimed farmland productivity should recover quickly and agriculture production would be larger than in the TR and TR(MOD) plans; 3) The total estimated net revenue in the DSR plan should be 2.8 times and 1.2 times more than that in the TR and TR (MOD) plans; and 4) The total net revenue of the TR(MOD) plan should be increased by 8.1% as compared with the TR plan. The above net revenue benefits should be much greater for analyses over longer periods. Furthermore, in DSR, availability of farmland and water resources will be much longer throughout the active mining period. The mining and reclaimed areas in DSR can therefore provide improved socio-economic environment through community business development and settlement of labor markets during and after mining ceases in the area. Although solutions and sample calculations presented are simple, the paper clearly demonstrates the usefulness of the DSR concepts to minimize negative impacts to the environment while enhancing long-term socio-economic environment.

The authors hope that this study will encourage mine operators to implement DSR concepts in planning while considering new and ongoing minerals development projects. DSR is a powerful engineering and planning concept that can minimize land, water [14,31], and air impacts of mining and enhance socio-economic conditions under a variety of mining conditions. Furthermore, it should not be limited to only reclamation planning. Mining and DSR planning should be integrated to develop both optimal mining and reclamation plans [10,13,39] to enhance profitability of mining venture and its sustainability. That should further develop new ideas for both mining and reclamation for improved sustainability of mineral projects. Toward this goal the authors recommend that every mining project, irrespective of the stage of its development, should consider having a steering committee consisting of mining, business, and regional development professionals to review alternate opportunities for mining and reclamation to improve project sustainability while maximizing profit potential. That would also ensure the development of sound mine closure plans [42] and slow implementation of the concept “mine closure must begin the day mining starts” while maximizing profit potential.

Reviewer 2 Report

I believe that the problem of the exploitation and use of coal is current and must be dealt with. It is very necessary that coal be progressively replaced by less polluting, healthier and more environmentally friendly forms of energy, such as wind or solar energy, in the context of climate change, which is so important. In addition, underground mining generates a lot of loss of human life.

The most important lack or weakness that I see in the paper presented is that the plans studied TR, TR (MOD) and DSR are not contrasted with other papers. There is no discussion, section 4 of the Discussion is not such a discussion, it is only about from the opinion of the authors. The paper needs to introduce references in the results and discussion, since there are none, not a single reference.

The paper only has 23 references that are included in the introduction and some in methods, but none are used to compare the authors' results, references are missing to discuss the authors' proposals.

The references are all from Chinese scientists, it would be interesting to introduce others from authors from other countries and make the proposal more global than Chinese, because the IJERPH journal has global repercussions and crosses all borders.

Therefore, the bibliography should be expanded for results and discussion and made more international, that is the greatest weakness of this work, from my point of view.

The bibliography does not adapt to the format of the journal, in which the year of each article should be in bold and the name of the journal in italics and abbreviated. It must conform to the rules.

Most of the tables go outside the margins of the text.

The figures are very illustrative and appropriate.

Author Response

Comments 1: The most important lack or weakness that I see in the paper presented is that the plans studied TR, TR (MOD) and DSR are not contrasted with other papers. There is no discussion, section 4 of the Discussion is not such a discussion, it is only about from the opinion of the authors. The paper needs to introduce references in the results and discussion, since there are none, not a single reference.

Responses and a summary of revisions:

Thanks for your suggestion and a very good comment. Discussion and Concluding Remarks section has been rewritten to be responsive to this comment with added references. See Line 463-559. In the results section, references have been added to compare results with previous research in Line 397, 406, 426, 459.

With significant continuing coal production from underground mining in China and global pressures for sustainable development, there has been a great need for increased research on reclaiming unstable subsided lands. Zhao [32] and Xiao [33] researched optimizing reclamation time but did not quantify the benefits economically. Zhang [37] advanced the above for mining multiple coal seams. Li et al. [10,13] and Feng et al. [39] worked to modify underground mine plans to minimize subsidence impacts but did not consider original surface topography in analysis. Chen and Hang [38] focused on soil reconstruction procedures. Hu et al. [31] developed DSR concepts for mining areas with shallow groundwater resources and incorporated economic analyses. However, they did not consider separating topsoil and subsoil, socio-economic development concepts of super-farm, large water storage, and associated small- and large-scale business development that are included in analyses here. So, the authors have incorporated several new concepts into DSR planning: 1) Creating super-farmland areas of high agricultural productivity to offset the loss of farmland getting submerged under water, and planting fruits and vegetables to enhance revenue streams; 2) Creating large areas of planned quality water resources to serve existing and new or improved socio-economic ventures to support long-term community needs at the local and regional levels, and creating new businesses of water sports and fish breeding areas; 3) Perform simple cost-benefit analyses; 4) In addition to TR develop a TR(MOD) plan to analyze the importance of stripping soils before they submerge into water and separating topsoil and subsoil; 5) Consider original surface topography to predict surface subsidence and develop DSR, TR and TR (MOD) reclamation plans based on dynamic subsidence topography.

This paper has attempted a conceptual implementation of DSR planning concepts at a large coal mine involving five single seam longwall faces in a mining area with shallow groundwater table about 1.5 m below the ground surface and 4.4 m of maximum surface subsidence due to mining. The study involved mining and post-mining reclamation to assess if DSR could have improved both the environment and socio-economic conditions for post-mining land use as compared to using TR approaches in China. In DSR topsoil, subsoil, farming, and water resources management were dynamically synergized concurrent with mining. Within TR, two approaches were considered: 1) Designated as TR, all reclamation activities are initiated only after all mining had been completed in the area and land was allowed to submerge into water in the subsiding areas; and 2) TR (MOD) where soils (topsoil and subsoil) ahead of mining, likely to be submerged, were stripped and stored without separation into topsoil and subsoil for reclamation after all mining was completed in the area. The authors undertook the current analyses for this hypothetical case as soon as the data became available.

A comparison of the three analyzed plans shows that: 1) Upon final reclamation, farmland area and water resources in the DSR and TR (MOD) plans are increased by 5.6% and 30.2% as compared to the TR plan. Stripping soils before they submerge into water is important for farmland reclamation and long-term economic development; 2) Due to topsoil and subsoil separation in DSR plan, reclaimed farmland productivity should recover quickly and agriculture production would be larger than in the TR and TR(MOD) plans; 3) The total estimated net revenue in the DSR plan should be 2.8 times and 1.2 times more than that in the TR and TR (MOD) plans; and 4) The total net revenue of the TR(MOD) plan should be increased by 8.1% as compared with the TR plan. The above net revenue benefits should be much greater for analyses over longer periods. Furthermore, in DSR, availability of farmland and water resources will be much longer throughout the active mining period. The mining and reclaimed areas in DSR can therefore provide improved socio-economic environment through community business development and settlement of labor markets during and after mining ceases in the area. Although solutions and sample calculations presented are simple, the paper clearly demonstrates the usefulness of the DSR concepts to minimize negative impacts to the environment while enhancing long-term socio-economic environment.

The authors hope that this study will encourage mine operators to implement DSR concepts in planning while considering new and ongoing minerals development projects. DSR is a powerful engineering and planning concept that can minimize land, water [14,31], and air impacts of mining and enhance socio-economic conditions under a variety of mining conditions. Furthermore, it should not be limited to only reclamation planning. Mining and DSR planning should be integrated to develop both optimal mining and reclamation plans [10,13,39] to enhance profitability of mining venture and its sustainability. That should further develop new ideas for both mining and reclamation for improved sustainability of mineral projects. Toward this goal the authors recommend that every mining project, irrespective of the stage of its development, should consider having a steering committee consisting of mining, business, and regional development professionals to review alternate opportunities for mining and reclamation to improve project sustainability while maximizing profit potential. That would also ensure the development of sound mine closure plans [42] and slow implementation of the concept “mine closure must begin the day mining starts” while maximizing profit potential.

Comments 2: The paper only has 23 references that are included in the introduction and some in methods, but none are used to compare the authors' results, references are missing to discuss the authors' proposals.

Responses and a summary of revisions:

Thanks for your suggestion. Additional references have been added in Discussion and Concluding Remarks to compare and contrast our results. See Line 463-559. In the Results part, references are also be added to contrast with previous research in Line 397, 406, 426, 459.

With significant continuing coal production from underground mining in China and global pressures for sustainable development, there has been a great need for increased research on reclaiming unstable subsided lands. Zhao [32] and Xiao [33] researched optimizing reclamation time but did not quantify the benefits economically. Zhang [37] advanced the above for mining multiple coal seams. Li et al. [10,13] and Feng et al. [39] worked to modify underground mine plans to minimize subsidence impacts but did not consider original surface topography in analysis. Chen and Hang [38] focused on soil reconstruction procedures. Hu et al. [31] developed DSR concepts for mining areas with shallow groundwater resources and incorporated economic analyses. However, they did not consider separating topsoil and subsoil, socio-economic development concepts of super-farm, large water storage, and associated small- and large-scale business development that are included in analyses here. So, the authors have incorporated several new concepts into DSR planning: 1) Creating super-farmland areas of high agricultural productivity to offset the loss of farmland getting submerged under water, and planting fruits and vegetables to enhance revenue streams; 2) Creating large areas of planned quality water resources to serve existing and new or improved socio-economic ventures to support long-term community needs at the local and regional levels, and creating new businesses of water sports and fish breeding areas; 3) Perform simple cost-benefit analyses; 4) In addition to TR develop a TR(MOD) plan to analyze the importance of stripping soils before they submerge into water and separating topsoil and subsoil; 5) Consider original surface topography to predict surface subsidence and develop DSR, TR and TR (MOD) reclamation plans based on dynamic subsidence topography.

This paper has attempted a conceptual implementation of DSR planning concepts at a large coal mine involving five single seam longwall faces in a mining area with shallow groundwater table about 1.5 m below the ground surface and 4.4 m of maximum surface subsidence due to mining. The study involved mining and post-mining reclamation to assess if DSR could have improved both the environment and socio-economic conditions for post-mining land use as compared to using TR approaches in China. In DSR topsoil, subsoil, farming, and water resources management were dynamically synergized concurrent with mining. Within TR, two approaches were considered: 1) Designated as TR, all reclamation activities are initiated only after all mining had been completed in the area and land was allowed to submerge into water in the subsiding areas; and 2) TR (MOD) where soils (topsoil and subsoil) ahead of mining, likely to be submerged, were stripped and stored without separation into topsoil and subsoil for reclamation after all mining was completed in the area. The authors undertook the current analyses for this hypothetical case as soon as the data became available.

A comparison of the three analyzed plans shows that: 1) Upon final reclamation, farmland area and water resources in the DSR and TR (MOD) plans are increased by 5.6% and 30.2% as compared to the TR plan. Stripping soils before they submerge into water is important for farmland reclamation and long-term economic development; 2) Due to topsoil and subsoil separation in DSR plan, reclaimed farmland productivity should recover quickly and agriculture production would be larger than in the TR and TR(MOD) plans; 3) The total estimated net revenue in the DSR plan should be 2.8 times and 1.2 times more than that in the TR and TR (MOD) plans; and 4) The total net revenue of the TR(MOD) plan should be increased by 8.1% as compared with the TR plan. The above net revenue benefits should be much greater for analyses over longer periods. Furthermore, in DSR, availability of farmland and water resources will be much longer throughout the active mining period. The mining and reclaimed areas in DSR can therefore provide improved socio-economic environment through community business development and settlement of labor markets during and after mining ceases in the area. Although solutions and sample calculations presented are simple, the paper clearly demonstrates the usefulness of the DSR concepts to minimize negative impacts to the environment while enhancing long-term socio-economic environment.

The authors hope that this study will encourage mine operators to implement DSR concepts in planning while considering new and ongoing minerals development projects. DSR is a powerful engineering and planning concept that can minimize land, water [14,31], and air impacts of mining and enhance socio-economic conditions under a variety of mining conditions. Furthermore, it should not be limited to only reclamation planning. Mining and DSR planning should be integrated to develop both optimal mining and reclamation plans [10,13,39] to enhance profitability of mining venture and its sustainability. That should further develop new ideas for both mining and reclamation for improved sustainability of mineral projects. Toward this goal the authors recommend that every mining project, irrespective of the stage of its development, should consider having a steering committee consisting of mining, business, and regional development professionals to review alternate opportunities for mining and reclamation to improve project sustainability while maximizing profit potential. That would also ensure the development of sound mine closure plans [42] and slow implementation of the concept “mine closure must begin the day mining starts” while maximizing profit potential.

Comments 3: The references are all from Chinese scientists, it would be interesting to introduce others from authors from other countries and make the proposal more global than Chinese, because the IJERPH journal has global repercussions and crosses all borders.

Responses and a summary of revisions:

Thanks for your suggestion. References have now been added in Introduction, Results, Discussion and Concluding Remarks sections, and most of these are international articles. See Line 37-121, 397, 406, 426, 459, 463-559 and Reference part Line 575-670.

Comments 4: Therefore, the bibliography should be expanded for results and discussion and made more international, that is the greatest weakness of this work, from my point of view.

Responses and a summary of revisions:

Thanks for your suggestion. Bibliography has been expanded for Results, Discussion and Concluding Remarks sections, and most of these are international articles. See Line 397, 406, 426, 459, 463-559 and Reference part Line 575-670.

With significant continuing coal production from underground mining in China and global pressures for sustainable development, there has been a great need for increased research on reclaiming unstable subsided lands. Zhao [32] and Xiao [33] researched optimizing reclamation time but did not quantify the benefits economically. Zhang [37] advanced the above for mining multiple coal seams. Li et al. [10,13] and Feng et al. [39] worked to modify underground mine plans to minimize subsidence impacts but did not consider original surface topography in analysis. Chen and Hang [38] focused on soil reconstruction procedures. Hu et al. [31] developed DSR concepts for mining areas with shallow groundwater resources and incorporated economic analyses. However, they did not consider separating topsoil and subsoil, socio-economic development concepts of super-farm, large water storage, and associated small- and large-scale business development that are included in analyses here. So, the authors have incorporated several new concepts into DSR planning: 1) Creating super-farmland areas of high agricultural productivity to offset the loss of farmland getting submerged under water, and planting fruits and vegetables to enhance revenue streams; 2) Creating large areas of planned quality water resources to serve existing and new or improved socio-economic ventures to support long-term community needs at the local and regional levels, and creating new businesses of water sports and fish breeding areas; 3) Perform simple cost-benefit analyses; 4) In addition to TR develop a TR(MOD) plan to analyze the importance of stripping soils before they submerge into water and separating topsoil and subsoil; 5) Consider original surface topography to predict surface subsidence and develop DSR, TR and TR (MOD) reclamation plans based on dynamic subsidence topography.

This paper has attempted a conceptual implementation of DSR planning concepts at a large coal mine involving five single seam longwall faces in a mining area with shallow groundwater table about 1.5 m below the ground surface and 4.4 m of maximum surface subsidence due to mining. The study involved mining and post-mining reclamation to assess if DSR could have improved both the environment and socio-economic conditions for post-mining land use as compared to using TR approaches in China. In DSR topsoil, subsoil, farming, and water resources management were dynamically synergized concurrent with mining. Within TR, two approaches were considered: 1) Designated as TR, all reclamation activities are initiated only after all mining had been completed in the area and land was allowed to submerge into water in the subsiding areas; and 2) TR (MOD) where soils (topsoil and subsoil) ahead of mining, likely to be submerged, were stripped and stored without separation into topsoil and subsoil for reclamation after all mining was completed in the area. The authors undertook the current analyses for this hypothetical case as soon as the data became available.

A comparison of the three analyzed plans shows that: 1) Upon final reclamation, farmland area and water resources in the DSR and TR (MOD) plans are increased by 5.6% and 30.2% as compared to the TR plan. Stripping soils before they submerge into water is important for farmland reclamation and long-term economic development; 2) Due to topsoil and subsoil separation in DSR plan, reclaimed farmland productivity should recover quickly and agriculture production would be larger than in the TR and TR(MOD) plans; 3) The total estimated net revenue in the DSR plan should be 2.8 times and 1.2 times more than that in the TR and TR (MOD) plans; and 4) The total net revenue of the TR(MOD) plan should be increased by 8.1% as compared with the TR plan. The above net revenue benefits should be much greater for analyses over longer periods. Furthermore, in DSR, availability of farmland and water resources will be much longer throughout the active mining period. The mining and reclaimed areas in DSR can therefore provide improved socio-economic environment through community business development and settlement of labor markets during and after mining ceases in the area. Although solutions and sample calculations presented are simple, the paper clearly demonstrates the usefulness of the DSR concepts to minimize negative impacts to the environment while enhancing long-term socio-economic environment.

The authors hope that this study will encourage mine operators to implement DSR concepts in planning while considering new and ongoing minerals development projects. DSR is a powerful engineering and planning concept that can minimize land, water [14,31], and air impacts of mining and enhance socio-economic conditions under a variety of mining conditions. Furthermore, it should not be limited to only reclamation planning. Mining and DSR planning should be integrated to develop both optimal mining and reclamation plans [10,13,39] to enhance profitability of mining venture and its sustainability. That should further develop new ideas for both mining and reclamation for improved sustainability of mineral projects. Toward this goal the authors recommend that every mining project, irrespective of the stage of its development, should consider having a steering committee consisting of mining, business, and regional development professionals to review alternate opportunities for mining and reclamation to improve project sustainability while maximizing profit potential. That would also ensure the development of sound mine closure plans [42] and slow implementation of the concept “mine closure must begin the day mining starts” while maximizing profit potential.

Comments 5: The bibliography does not adapt to the format of the journal, in which the year of each article should be in bold and the name of the journal in italics and abbreviated. It must conform to the rules.

Responses and a summary of revisions:

Thanks for your comment. All references have now been modified and follow journal guidelines. See Line 575-670.

Comments 6: Most of the tables go outside the margins of the text.

Responses and a summary of revisions:

Thanks for your comment. Table widths have been decreased for Table 3 in Line 215, Table 4 in Line 382, Table 5 in Line 394, Table 6 in Line 401 and Table 7 in Line 423.

Comments 7: The figures are very illustrative and appropriate.

Responses and a summary of revisions:

We appreciate your comment.

Round 2

Reviewer 2 Report

The authors have responded to all my suggestions and I believe that the paper has improved substantially. Especially they have introduced more references in the discussion section and it has been enriched by increasing the references consulted.

Author Response

Comments 1: The authors have responded to all my suggestions and I believe that the paper has improved substantially. Especially they have introduced more references in the discussion section and it has been enriched by increasing the references consulted.

Responses:

We appreciate your comment.